# A Study on the Power Reserve of Distributed Generators Based on Power Sensitivity Analysis in a Large-Scale Power System

Dongmin Kim [1], Jung-Wook Park [1,*] and Soo Hyoung Lee [2,*]

1    School of Electrical & Electronic Engineering, Yonsei University, Seoul 03722, Korea; ys0641056@yonsei.ac.kr
2    Department of Electrical and Control Engineering, Mokpo National University, Mokpo 58554, Korea
*    Correspondence: jungpark@yonsei.ac.kr (J.-W.P.); slee82@mokpo.ac.kr (S.H.L.)

**Abstract:** Converter-based generators (CBGs) that use renewable energy sources (RESs) are replacing traditional aging coal and nuclear power generators. Increasing the penetration of CBGs into the entire power generation process reduces both the inertia constant of the power system and the total amount of power reserves. Additionally, RESs are very intermittent and it is difficult to predict changes in them. These problems, due to CBGs using RESs, pose new challenges to net–load balancing. As a solution, this paper proposes a virtual multi-slack (VMS) droop control that secures the stability and efficiency of system operation by controlling the output of CBGs distributed in various regions. The VMS droop control makes it possible to increase the inertia constant of the power system and to respond quickly and appropriately to load changes through the proposed VMS droop control based on power sensitivity. It is also proposed that the process selects proper power reserves of CBGs for stable VMS droop control. To verify the effectiveness of the proposed VMS droop control and the proper power reserve selection method for CBGs, several case studies were performed using a real Korean power system.

**Keywords:** converter-based generator; high renewable penetration; large-scale power system; power sensitivity analysis; virtual multi-slack droop control

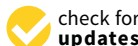



## 1. Introduction

Coal power plants, which account for the largest proportion of Korea's total power generation at 40%, cause environmental pollution problems by emitting dust and wastewater. In the case of nuclear power plants, meanwhile, which account for the second-largest proportion of total power generation at 30%, they cause a rise in social concerns due to frequent earthquakes in their areas [1]. To deal with these environmental and social safety issues in power generation, the huge penetration of converter-based generators (CBGs) using renewable energy sources (RESs) is being increasingly considered [2]. CBGs using RESs are environmentally friendly and safe compared to conventional synchronous generators (SGs). As a part of this effort, the Korean government announced the "Renewable Energy 3020 Action Plan" in 2017, which aims to increase the total power generation dependency on RESs by up to 20% of the national total power generation by 2030. According to this national energy plan, many conventional aging coal and nuclear power plants are to be replaced by CBGs using RESs (mainly wind turbine (WT) and photovoltaic (PV) generators, as shown in Table 1). As a result, the facility capacity of CBGs will increase to 63.8 GW by 2030, and large-scale wind plants and PV plants will be newly installed in Areas 1, 2, and 3, as shown in Figure 1. While 40% of the total load is concentrated in the metro area, the large-scale CBGs that are to be newly installed will be distributed in different areas. The stability of the power system must be solved along with the social acceptance of the expansion of large-scale RESs [3].

**Table 1.** Renewable energy capacity expansion plan in Korea by 2030 (MW).

| Year | PV | Wind | Hydro | Offshore | Bio | Wastes | Byproduct Gas | Fuel Cell | IGCC | Total |
|------|------|------|------|------|------|------|------|------|------|------|
| 2017 | 5030 | 1174 | 1795 | 255 | 725 | 323 | 1377 | 291 | 346 | 11,316 |
| 2020 | 9330 | 2724 | 1850 | 255 | 1025 | 323 | 1377 | 531 | 346 | 17,761 |
| 2025 | 19,530 | 8474 | 1960 | 255 | 1405 | 323 | 1377 | 691 | 746 | 34,761 |
| 2030 | 33,530 | 17,674 | 2105 | 255 | 1705 | 323 | 1377 | 746 | 746 | 58,461 |

PV, photovoltaic; IGCC, integrated gasification combined cycle.

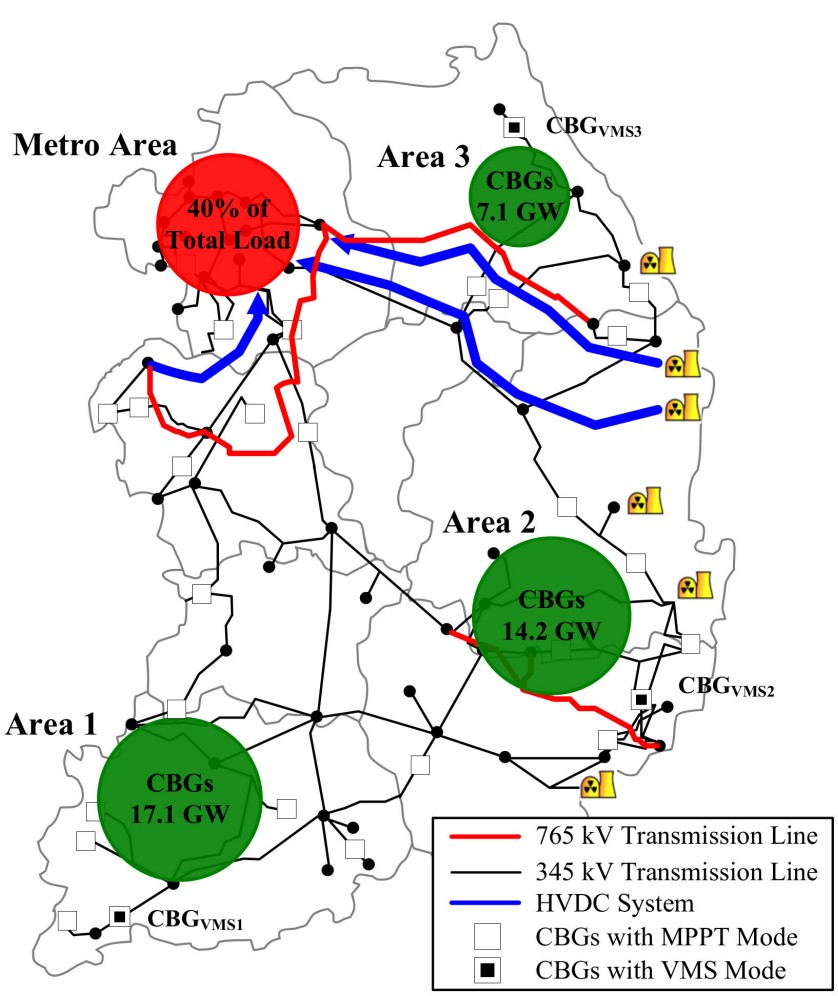

**Figure 1.** The Korean 2030 power system plan with high renewable penetration. CBGs, converter-based generators; MPPT, maximum power point tracking; VMS, virtual multi-slack.

The output power variations of the CBGs using RESs are significantly affected by the external environment. Therefore, the uncertainty and intermittency of RESs make it difficult to balance the total generation and load consumption of the power system [4–6]. This imbalance not only causes unstable operational problems in the power system due to sudden changes in the output of CBGs, but also causes problems such as increased congestion of lines due to the excessive power of CBGs. Moreover, replacing the existing synchronous generators (SGs) with large-scale CBGs reduces the total inertia of the power system [7]. Low inertia of the power system causes a large frequency change even in small imbalances between power generation and load consumption. As such, the penetrations of CBGs are limited by the deterioration of the stability, reliability, and quality of the power system resulting from CBGs.

To increase the penetration of CBGs while overcoming those problems, many research papers have been published on power system stabilization techniques using PV and wind turbine generators (WTGs) and operation strategies of energy storage systems (ESSs). When the load increases and the generation is insufficient, techniques for increasing the output power of CBGs by limiting the PV output power and controlling the pitch angle and rotor speed of WTGs have been proposed [8–13]. This ensures the stability of the power system by providing additional power reserves to the power system similarly to conventional SGs. Additionally, there are inertial response techniques and droop control techniques for CBGs, and these control methods allow the output power of the CBGs to be properly controlled in response to the power system status. For more stable and efficient power management of CBGs, several strategies for ESSs connected to CBGs have been proposed [14–16]. By using the fast and efficient charging and discharging features of ESSs, it is possible to offset the output power changes of CBGs and to increase the efficiency of power use by storing the remaining output power of CBGs.

This paper proposes a virtual multi-slack (VMS) droop control based on power sensitivity analysis. Using this control, the actual slack (one SG for the actual slack) and the virtual slacks (several CBGs operating in the VMS droop control) share the charge of clearing the power imbalance between the generation and the load consumption [17]. It is reasonable to share the charge of clearing the power mismatch between representative CBGs, rather than all SGs and CBGs. This is because the mechanical parts of SGs can be stressed by the variable power, and the independent power producer (IPP)-owned CBGs are not controllable by the utility. To resolve the power imbalance by adjusting the outputs of several CBGs, each CBG must have a sufficient reserve to increase the output power. Especially for the power system operator, stable system operation needs to secure sufficient power reserves. On the contrary, it is always important for CBG operators to keep the output of the CBG as high as possible; therefore, it is essential to calculate the proper power reserve of each CBG. In this paper, the appropriate power reserve of each CBG was calculated according to the condition of the power system based on power sensitivity.

This paper is organized as follows: Section 2 describes the VMS power flow analysis and the basic concept of the proposed VMS droop control based on power sensitivity. In Section 3, the method for calculating the appropriate power reserve of CBGs is presented. Thereafter, case studies are carried out on the large-scale power system with high renewable penetration in Section 4. Finally, the conclusions are given in Section 5.

## 2. VMS Power Flow Analysis

### 2.1. Concept of the Proposed Power System Operation with VMS Droop Control

In the past, several small-scale CBGs using RESs have been connected to the power system based on the SGs in various regions. In this conventional power system, stable power system operation is possible based on the response of the SGs that are robust to system changes. For this reason, the main concern of the operation of CBGs is not the stable operation of the power system, but the way in which to maintain the maximum output power and to increase its amount of maximum output power. However, as large-scale CBGs using the RESs replace conventional SGs, the penetration of these SGs in the whole power system decreases. As a result, stability problems for power system operation have arisen. The proposed VMS droop control based on power sensitivity analysis improves power system stability by quickly adjusting the output power of each CBG responding to power system changes. Additionally, it is necessary to sufficiently increase the responding abilities of CBGs by securing an appropriate power reserve in each CBG as in conventional SGs. Figure 2 shows conventional CBGs under maximum power point tracking (MPPT) with a fixed maximum power and the proposed CBGs that can increase the power with the power reserve based on VMS.

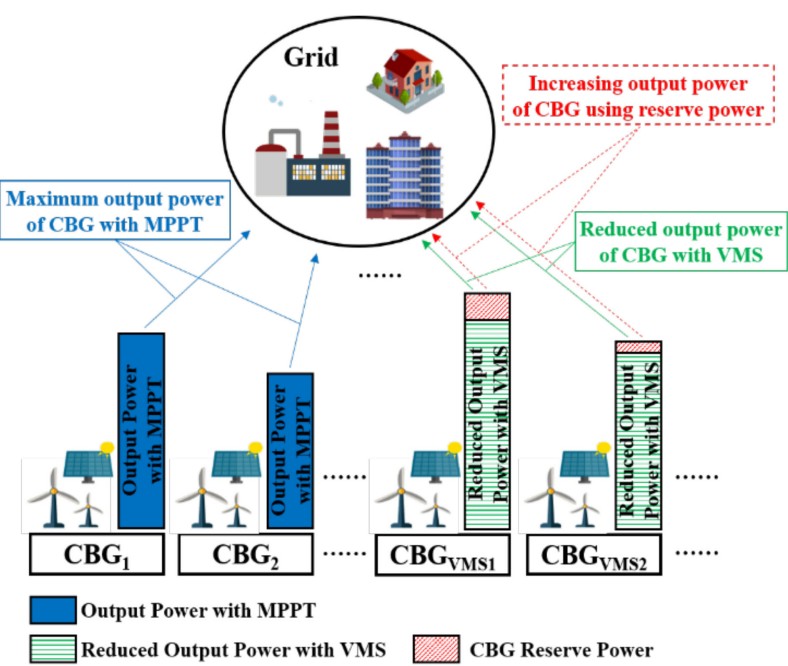

**Figure 2.** Power system based on the proposed VMS power system including CBGs with power reserve.

*2.2. VMS Power Flow Analysis*

The conventional electric power grid has only one slack bus (actual slack bus), which has a specified magnitude and phase angle of voltage. In the proposed VMS droop control, both the actual and the virtual slack buses can operate as slacks, and they can compensate for the difference between generation and load consumption. Generally, the real and reactive power mismatches in the $n$ buses system are given as:

$$\Delta P_i = \Delta P_{i,VMS} + \Delta P_{i,Net} = P_i - \sum_{j=1}^{n} |V_i||V_j||Y_{ij}| \cos(\theta_{ij} - \delta_i + \delta_j) \tag{1}$$

$$\Delta Q_i = \Delta Q_{i,VMS} + \Delta Q_{i,Net} = Q_i + \sum_{j=1}^{n} |V_i||V_j||Y_{ij}| \sin(\theta_{ij} - \delta_i + \delta_j) \tag{2}$$

where $P_i$ and $Q_i$ are the real power and reactive power of bus $i$, respectively [18]; $P_i$ ($Q_i$) consists of $P_{i,VMS}$ ($Q_{i,VMS}$) and $P_{i,Net}$ ($Q_{i,Net}$); $P_{i,VMS}$ ($Q_{i,VMS}$) is the real (reactive) power from the CBGs connected to bus $i$ under the VMS droop control; $P_{i,Net}$ ($Q_{i,Net}$) is the sum of the real (reactive) power of the other generators and loads of bus $i$.

$$\begin{bmatrix} \Delta\delta_2 \\ \vdots \\ \Delta\delta_m \\ \vdots \\ \Delta\delta_n \\ \hline \Delta V_2 \\ \vdots \\ \Delta V_m \\ \vdots \\ \Delta V_n \end{bmatrix} = \begin{bmatrix} J_{P\delta} & J_{PV} \\ J_{Q\delta} & J_{QV} \end{bmatrix}^{-1} \begin{bmatrix} \Delta P_{2,VMS} + \Delta P_{2,Net} \\ \vdots \\ \Delta P_{m,VMS} + \Delta P_{m,Net} \\ \vdots \\ \Delta P_{n,Net} \\ \hline \Delta Q_{2,VMS} + \Delta Q_{2,Net} \\ \vdots \\ \Delta Q_{m,VMS} + \Delta Q_{m,Net} \\ \vdots \\ \Delta Q_{n,Net} \end{bmatrix}, \quad K = \begin{bmatrix} K_{11} & K_{12} \\ K_{21} & K_{22} \end{bmatrix} = \begin{bmatrix} J_{P\delta} & J_{PV} \\ J_{Q\delta} & J_{QV} \end{bmatrix}^{-1} \tag{3}$$

where $[\Delta\delta|\Delta V]^t$ and $[\Delta P|\Delta Q]^t$ are the vectors of voltage and power mismatches, respectively. The subscripts $n$ and $m$ represent the numbers of entire and slack buses ($m - 1$

virtual slacks and one actual slack), respectively. The inverse matrix of the Jacobian matrix, $\mathbf{J}$ ($\in \mathbb{R}^{2(n-1) \times 2(n-1)}$), is defined as $\mathbf{K}$. Then, the voltage and power mismatches in only the virtual slack buses can be calculated as:

$$
\begin{bmatrix} \Delta\delta_2 \\ \vdots \\ \Delta\delta_m \\ \hline \Delta V_2 \\ \vdots \\ \Delta V_m \end{bmatrix} = \mathbf{K}^{VMS} \begin{bmatrix} \Delta P_{2,VMS} \\ \vdots \\ \Delta P_{m,VMS} \\ \hline \Delta Q_{2,VMS} \\ \vdots \\ \Delta Q_{m,VMS} \end{bmatrix} + \mathbf{K}^{Net} \begin{bmatrix} \Delta P_{2,Net} \\ \vdots \\ \Delta P_{m,Net} \\ \vdots \\ \Delta P_{n,Net} \\ \hline \Delta Q_{2,Net} \\ \vdots \\ \Delta Q_{m,Net} \\ \vdots \\ \Delta Q_{n,Net} \end{bmatrix} \tag{4}
$$

where $\mathbf{K}^{VMS}$ ($\in \mathbb{R}^{2(m-1) \times 2(m-1)}$) and $\mathbf{K}^{Net}$ ($\in \mathbb{R}^{2(m-1) \times 2(n-1)}$) are the matrices reassigned as part of $\mathbf{K}$ in (3). As shown in (5), $\mathbf{K}^{VMS}$ includes the data of only the bases to which the CBGs under the VMS droop control are connected. $\mathbf{K}^{Net}$ includes the data of all buses except the actual slack bus and is given as (6).

$$
\mathbf{K}^{VMS} = \begin{bmatrix} K_{11}(2,2) & \cdots & K_{11}(2,m) & K_{12}(2,2) & \cdots & K_{12}(2,m) \\ \vdots & \ddots & \vdots & \vdots & \ddots & \vdots \\ K_{11}(m,2) & \cdots & K_{11}(m,m) & K_{12}(m,2) & \cdots & K_{12}(m,m) \\ \hline K_{21}(2,2) & \cdots & K_{21}(2,m) & K_{22}(2,2) & \cdots & K_{22}(2,m) \\ \vdots & \ddots & \vdots & \vdots & \ddots & \vdots \\ K_{21}(m,2) & \cdots & K_{21}(m,m) & K_{22}(m,2) & \cdots & K_{22}(m,m) \end{bmatrix} \tag{5}
$$

$$
\mathbf{K}^{Net} = \begin{bmatrix} K_{11}(2,2) & \cdots & K_{11}(2,n) & K_{12}(2,2) & \cdots & K_{12}(2,n) \\ \vdots & \ddots & \vdots & \vdots & \ddots & \vdots \\ K_{11}(m,2) & \cdots & K_{11}(m,n) & K_{12}(m,2) & \cdots & K_{12}(m,n) \\ \hline K_{21}(2,2) & \cdots & K_{21}(2,n) & K_{22}(2,2) & \cdots & K_{22}(2,n) \\ \vdots & \ddots & \vdots & \vdots & \ddots & \vdots \\ K_{21}(m,2) & \cdots & K_{21}(m,n) & K_{22}(m,2) & \cdots & K_{22}(m,n) \end{bmatrix} \tag{6}
$$

In the slack buses, including the virtual slack buses, the voltage magnitudes and phase angles are ideally specified. As a result, the mismatches of its voltages and phase angles are zero, meaning that the left side of (4) is zero. Therefore, (4) is reorganized as:

$$
\begin{bmatrix} \Delta P_{2,VMS} \\ \vdots \\ \Delta P_{m,VMS} \\ \hline \Delta Q_{2,VMS} \\ \vdots \\ \Delta Q_{m,VMS} \end{bmatrix} = -\left[\mathbf{K}^{VMS}\right]^{-1}\mathbf{K}^{Net} \begin{bmatrix} \Delta P_{2,Net} \\ \vdots \\ \Delta P_{m,Net} \\ \vdots \\ \Delta P_{n,Net} \\ \hline \Delta Q_{2,Net} \\ \vdots \\ \Delta Q_{m,Net} \\ \vdots \\ \Delta Q_{n,Net} \end{bmatrix} = \mathbf{S} \begin{bmatrix} \Delta P_{2,Net} \\ \vdots \\ \Delta P_{m,Net} \\ \vdots \\ \Delta P_{n,Net} \\ \hline \Delta Q_{2,Net} \\ \vdots \\ \Delta Q_{m,Net} \\ \vdots \\ \Delta Q_{n,Net} \end{bmatrix}, \ \mathbf{S} = -\left[\mathbf{K}^{VMS}\right]^{-1}\mathbf{K}^{Net} \tag{7}
$$

Through (7), the relationship between the power changes in specific buses and the changes in the power of the CBGs under VMS droop control can be related by using **S**, the power sensitivity matrix. When the power sensitivity matrix, **S**, is calculated, it is possible to determine the power changes of the CBGs corresponding to specific load changes through a simple calculation.

### 3. Calculating the Power Reserve for Each CBG

For a large-scale CBG to increase output power through VMS droop control, it must also have a certain level of power reserve, just as a conventional SG has a power reserve. When the load connected to a specific bus $i$ is increased, the proper amount of increase in the output power of each CBG can be calculated by (7). In a specific bus $i$, the change of only the load is given as:

$$
\mathbf{Ld}_i =
\begin{bmatrix}
0 \\
\vdots \\
\Delta P_{i,Load} \\
\vdots \\
0 \\
\hline
0 \\
\vdots \\
\Delta Q_{i,Load} \\
\vdots \\
0
\end{bmatrix}
\tag{8}
$$

where $\mathbf{Ld}_i$ is the load change at bus $i$. Using (7), the power reserve of each CBG required for the changes of specific loads can be expressed as:

$$
\begin{bmatrix}
P_{2,VMS} \\
\vdots \\
P_{m,VMS} \\
\hline
Q_{2,VMS} \\
\vdots \\
Q_{m,VMS}
\end{bmatrix}_{Reserve}
= \sum_{j=1}^{n} \mathbf{S} \cdot \mathbf{Ld}_j
\tag{9}
$$

In this paper, the power reserve of a CBG is largely classified as $P_{HS}^{res}$, $P_{HL}^{res}$, and $P_{CL}^{res}$ to calculate the appropriate power reserve that each CBG must possess. $P_{HS}^{res}$ is the power reserve that responds to increases in a hundred representative loads that have the highest power sensitivities with the CBG. $P_{HL}^{res}$ is the power reserve that responds to simultaneous increases of 10% of all loads that are greater than 100 MW. Finally, $P_{CL}^{res}$ is the power reserve that responds to load increases in the CBG-connected bus, where the amount that the load increases by is 10% of the largest load in the entire power system. Under normal conditions, the power reserve that a specific CBG should hold is the sum of $P_{HS}^{res}$ and $P_{HL}^{res}$. $P_{CL}^{res}$ is the power reserve for an extraordinary situation. Considering this, the power reserve that a specific CBG should possess is as follows:

$$
P^{res} = \max\{(P_{HS}^{res} + P_{HL}^{res}), P_{CL}^{res}\}
\tag{10}
$$

Finally, the initial output power of each CBG for operation with VMS droop control based on power sensitivity is as follows:

$$
P^{ini} = P^{MPPT} - P^{res}
\tag{11}
$$

where $P^{ini}$ and $P^{MPPT}$ are the initial power generation and the maximum power of the specific CBG, respectively.

## 4. Simulation Results

This section is composed of three parts to verify the VMS droop control effect and the adequacy of the power reserves. Section 4.1 shows the load changes in the representative buses. Section 4.2 describes the changes in the output powers of CBGs, the power system frequency, and the changes in the output powers of conventional power generators based on the conditions of Section 4.1. Finally, Section 4.3 shows the selection of the appropriate power reserve of CBGs based on power sensitivity analysis and verifies the results.

### 4.1. The Load Changes in the Representative Bus

Through the proposed power sensitivity-based VMS droop control, the distributed CBGs can quickly respond to load changes. To verify this, three representative CBGs were selected from Areas 1, 2, and 3, respectively. The three different control modes were compared using the three individual case studies in Section 4.2. In each case study, the same control mode was applied to all of the three CBGs simultaneously.

The load change was applied to the maximum load bus in the Metro Area in Figure 1 and the selected representative CBG buses. The amount the load changed by was 10% of the nation's single maximum load, and the load change increased sequentially in the Metro Area, and Areas 1, 2, and 3 every 60–240 s and then decreased again, as shown in Figure 3.

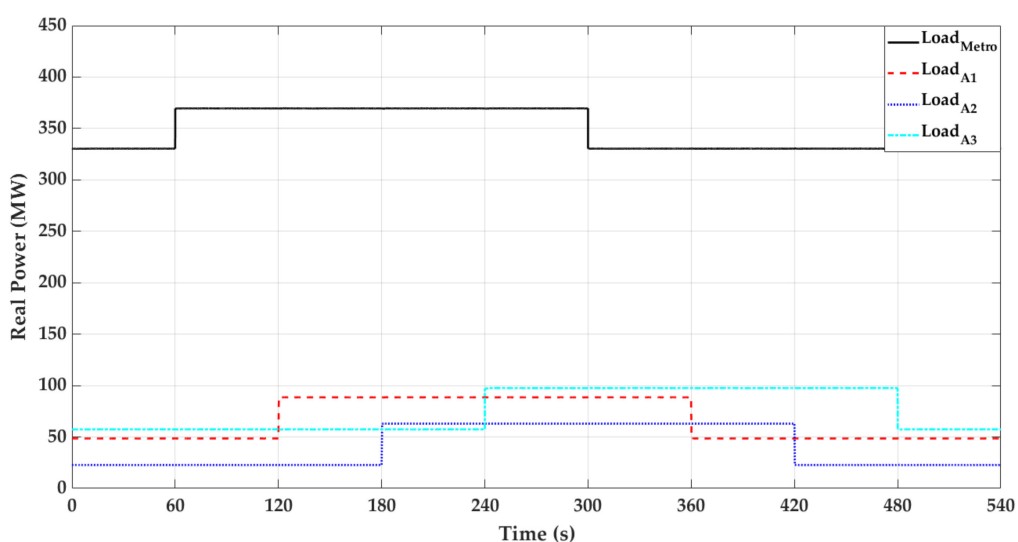

**Figure 3.** The changes in the representative loads.

### 4.2. Case Study 1: Results Based on the CBG Control Mode Differences

Three different control modes were compared using the three individual case studies in Figure 4. In each case study, the same control mode was applied to the three CBGs simultaneously. The first control mode was the MPPT mode, and the output power of the CBG always maintained the maximum output power regardless of the system situation. The second control mode was the proposed VMS droop control mode, which assumed an initial state with a certain amount of power reserve to cope with the load changes. Finally, the third control mode was the balanced response (BR) mode, in which all representative CBGs equally respond to the system change. The initial state was assumed to have the same power reserve in the BR mode as in the VMS droop control mode.

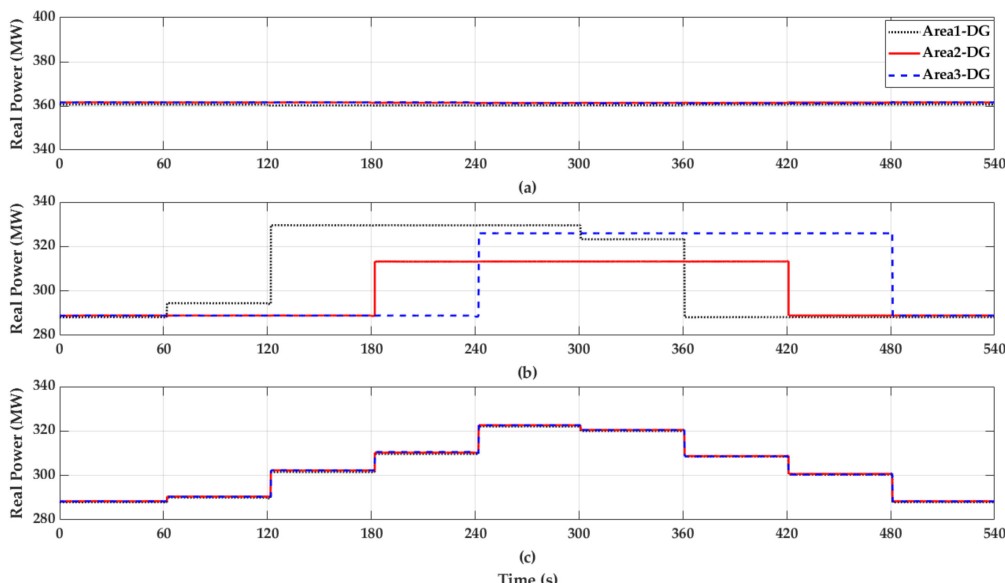

**Figure 4.** Real power responses of the representative CBGs with (**a**) MPPT, (**b**) VMS droop control, and (**c**) balanced response (BR).

The changes in the output powers of the representative CBGs corresponding to the load changes assumed in Section 4.1 can be seen in Figure 4. Figure 4a shows the simulation results of the CBGs in the MPPT control mode, and the output power is always fixed at the maximum power, even with all load changes. Figure 4b,c show the changes of the representative CBGs' output power according to the VMS droop control and BR modes, in which the initial generation amount was less than the initial generation amount in the MPPT control mode because of the same power reserve. In Figure 4b, where the representative CBGs respond with VMS droop control based on power sensitivity, the changes in the output powers corresponding to each representative CBG differ according to the region where the load change occurred. For example, only the representative CBG in Area 1 responds to the metro load increasing (decreasing) at 60 s (360 s).

Figure 5 shows the amount of increase or decrease in the power generation of the other generators, except for the representative CBGs. Figure 5a–c indicate the output power changes of all of the generators except for the representative CBGs located in Areas 1, 2, and 3, respectively. Figure 5d shows the output power changes of all of the generators in the rest of the region including the Metro Area. In the MPPT control mode, the change in the amount of output power of the other generators was the largest, and the generators in all regions operated regardless of the position of the load change. Thus, through the proposed VMS droop control mode, it is possible to efficiently reduce the total level of responses of the other generators when the amount of the load changes at a specific region. It might even be possible to reduce the mechanical stress of SGs by reducing the mechanical operation of said SGs.

Figure 6 shows the changes of total power system loss on real power and reactive power. The operation with BR is the most efficient in terms of reactive power, but this is not a significant difference considering the scale of the power system whose total loads are 50.445 GW and 10.990 GVAR.

Finally, Figure 7 shows the system frequency changes according to the control mode of the representative CBGs. The frequency nadir was the lowest in the MPPT control mode, in which the representative CBGs did not respond to the power system changes. In the proposed VMS droop control mode, the frequency fluctuation was the smallest, which means that this system can operate most stably.

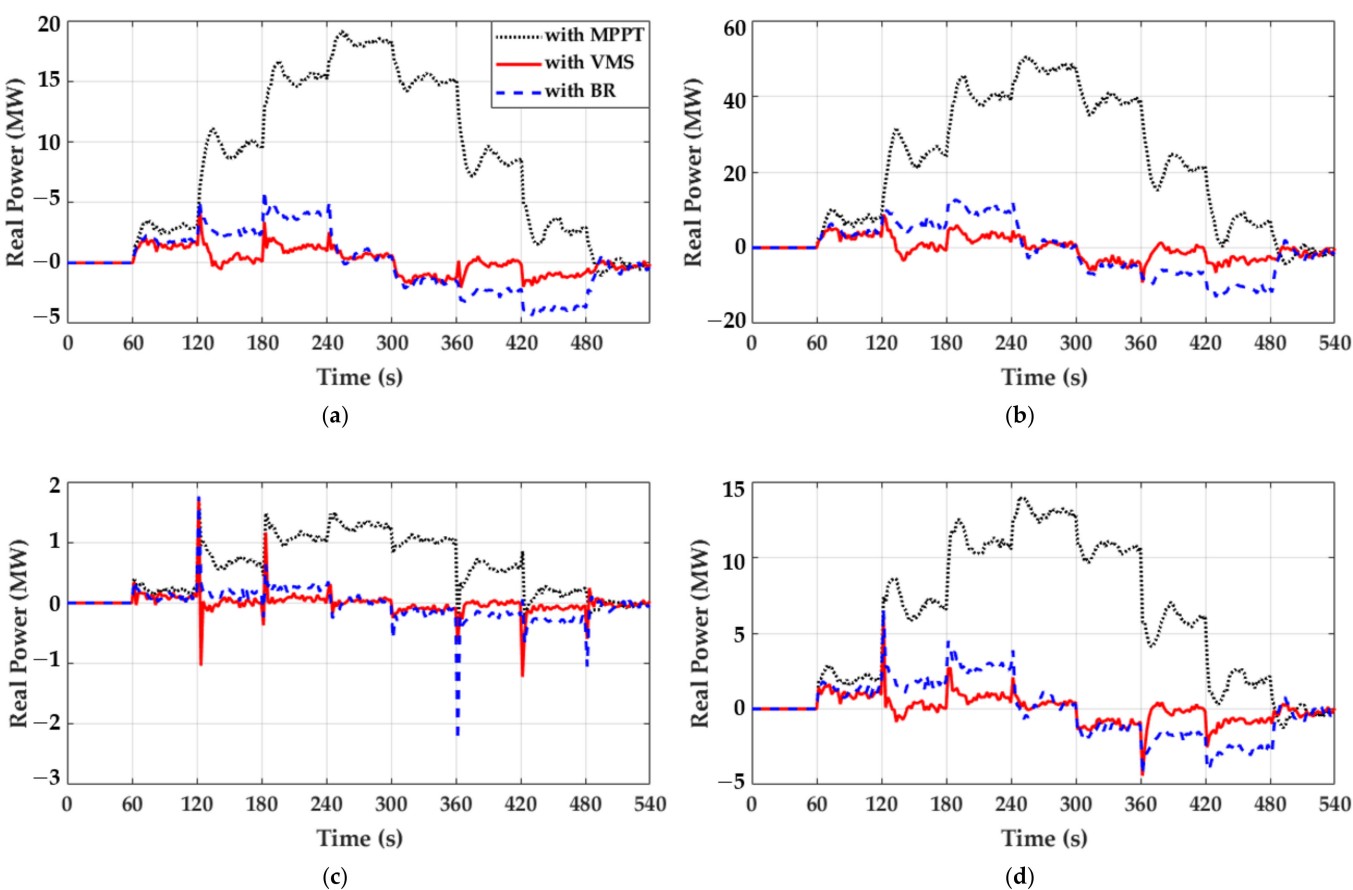

**Figure 5.** Total real power changes in (**a**) Area 1, (**b**) Area 2, (**c**) Area 3, and (**d**) the other areas, including Metro Area.

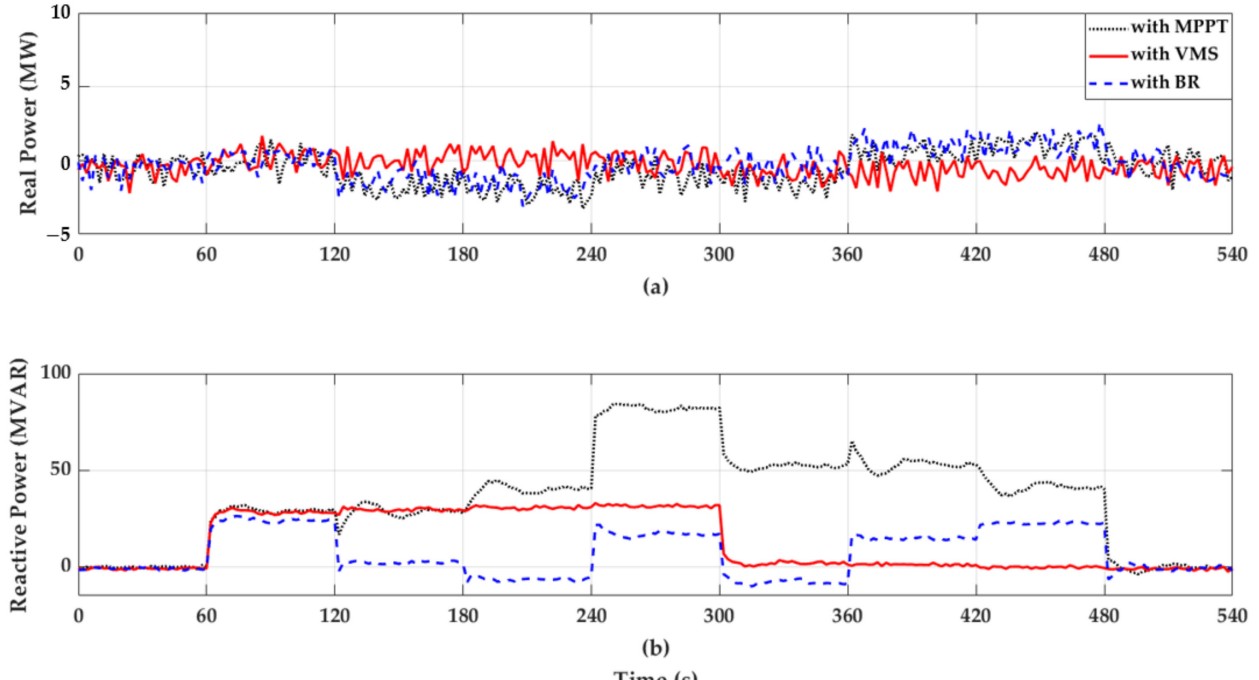

**Figure 6.** Total real (**a**) and reactive (**b**) power loss changes of the power system according to the CBG's control mode.

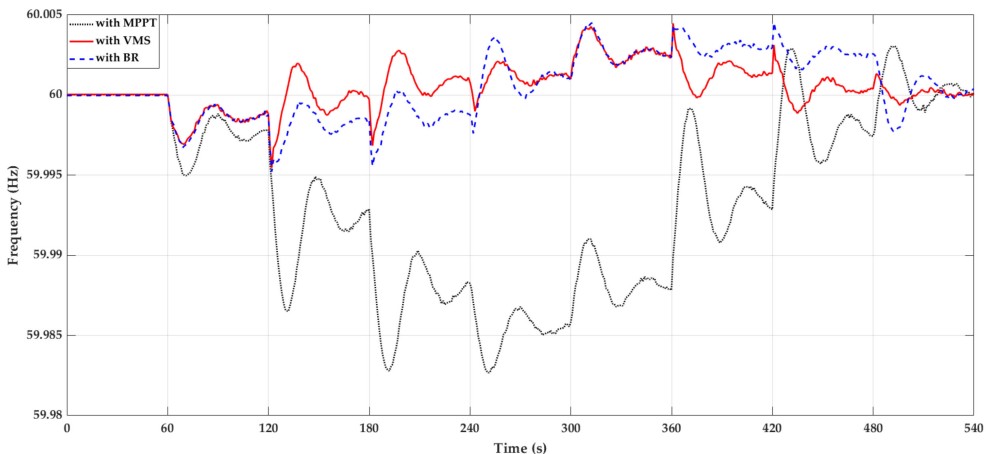

**Figure 7.** The frequency responses according to the CBG's control mode.

*4.3. Case Study 2: Results of the Difference in the Amount of Power Reserves of CBGs*

Another important problem is calculating an appropriate power reserve for stable power system operation through VMS droop control. The large power reserves of CBGs help to ensure the stability of the power system, but it is important for CBG operators to maintain maximum power for economic reasons. Based on power sensitivity analysis, the appropriate reserve power of each CBG can be determined, as shown in (10), and there is an advantage of being able to quickly determine an appropriate power reserve according to the power system situation through a simple calculation.

Table 2 shows the appropriate power reserve and the resulting initial output power of representative CBG by calculated through (8) to (11). The initial maximum output power ($P^{MPPT}$) of each CBG was assumed to be 360 MW. Although connected to the same power system, the appropriate power reserve of each CBG varied depending on the location where the CBG was connected. In particular, Area 1 had only 2 loads above 100 MW, while Areas 2 and 3 had 6 and 15, respectively. As a result, $P^{res}_{HL}$ in Area 1 was calculated to be much smaller than that in the other areas. In Area 1, the sum of $P^{res}_{HS}$ and $P^{res}_{HL}$ was less than $P^{res}_{CL}$, so $P^{res}$ was 50.1 MW, which was $P^{res}_{CL}$, and $P^{ini}$ was 309.9 MW. This initial output power was 86.08% of the initial maximum output power (360 MW). In Areas 2 and 3, $P^{res}$ was the sum of $P^{res}_{HS}$ and $P^{res}_{HL}$, which were 62.6 MW and 76.5 MW, respectively. As a result, $P^{ini}$ of Areas 2 and 3 was 297.4 MW (82.61%) and 283.5 MW (78.75%), respectively.

**Table 2.** Power reserve analysis for each CBG (MW).

|  | $P^{res}_{HS}$ | $P^{res}_{HL}$ | $P^{res}_{CL}$ | $P^{res}$ | $P^{ini}$ |
|---|---|---|---|---|---|
| $CBG_{A1}$ | 8.7 | 9 | 50.1 | 50.1 | 309.9 |
| $CBG_{A2}$ | 12.6 | 50 | 36.3 | 62.6 | 297.4 |
| $CBG_{A3}$ | 7.2 | 69.3 | 39 | 76.5 | 283.5 |

Figure 8 shows the changes in output power when the power reserves calculated in Table 2 are applied to each CBG (VMS with appropriate power reserve (APR)) and when all the power reserves of CBGs are fixed amounts (VMS with fixed power reserve (FPR)). The total amount of the power reserves of all CBGs under VMS was equal to 179 MW in both cases. In the case of VMS with APR, the power reserves assigned to each CBG are outlined in Table 2. In the case of VMS with FPR, the power reserves assigned to each CBG were fixed to 59.67 MW, one-third of the total power reserve. As a result, the initial power of each CBG was 300.33 MW, which is demonstrated by the black dotted line in Figure 8. The initial power of $CBG_{A1}$ in FPR was lower than that of APR and the initial powers of $CBG_{A2}$ and $CBG_{A3}$ in FPR were higher than those of APR.

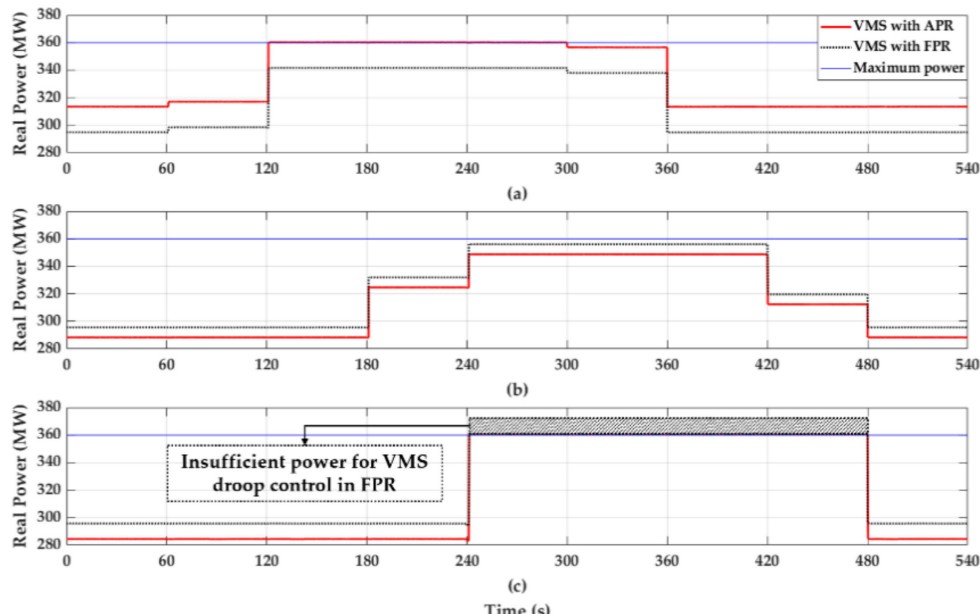

**Figure 8.** The comparison of output power of CBG according to its power reserve: (**a**) Area 1 (**b**) Area 2, and (**c**) Area 3. APR, appropriate power reserve; FPR, fixed power reserve.

To verify the appropriateness of the calculated power reserves, 10% increases (decreases) in representative loads in Area 3 were added for 240 s (480 s) in Case Study 1. As a result, the additional power reserves required in 240 s were 24 MW and 37.5 MW in $CBG_{A2}$ and $CBG_{A3}$, respectively. In FPR, the additional power from $CBG_{A3}$ was limited to 26.4 MW because of the limited size of the CBG.

Finally, Figure 9 shows the system frequency changes according to the different methods in the power reserve allocation of the representative CBGs. Before 240 s, there was no significant difference in frequency in either case. In the FPR, the lack of a power reserve of $CBG_{A3}$ greatly reduced frequency stability at 240 s.

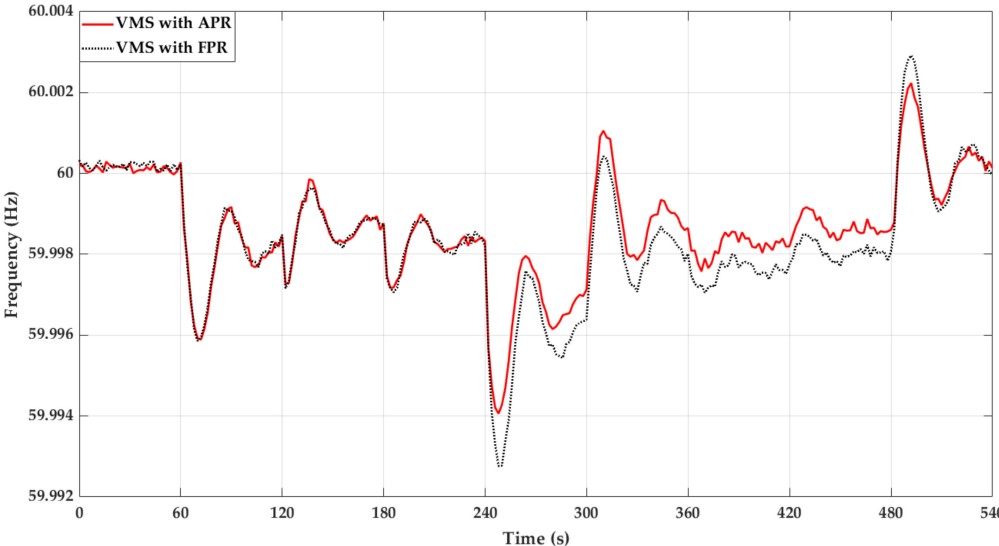

**Figure 9.** The frequency responses according to the power reserve allocation methods of CBGs.

## 5. Conclusions

This paper validated the improvement of power system stability through large-scale CBGs using VMS droop control. In order to secure the proposed operations of CBGs,

Korean power systems with real data were used in the simulation. The power sensitivity and power reserves were calculated with these practical Korean power systems. The output power of a CBG can be changed faster compared to that of conventional SGs in immediate response to the converter control signal. In addition, it is possible to quickly respond to load changes in various regions by adjusting the output powers of CBGs based on power sensitivity analysis. As a result, it can also reduce the mechanical stress in conventional SGs by minimizing the number of output power changes in the SGs.

In addition, an appropriate power reserve was calculated so that the CBGs responded to various load changes. By using the power sensitivity analysis to calculate the power reserve of CBGs, it was possible to calculate a more suitable power reserve for each CBG according to the power system's situation rather than the fixed amount of power reserves. The proposed power system stabilization control of CBGs can be applied to the current installation environment of CBGs by controlling the output of WTGs with applying pitch angle control or utilizing existing ESSs.

In the case of using ESS, there is an advantage that it can store the remaining electric power due to the power reserves and use it for mitigating the power mismatches or reducing the peak power regardless of the type of RESs. For this, studies on the integrated operation between ESSs and CBGs and the optimal capacity of ESSs are additionally needed. To apply the proposed control for the CBGs, the real-time monitoring of the whole power system using PMU is essential, and fast control for CBGs that can immediately respond to power system changes must be additionally studied.

**Author Contributions:** This research was conducted with the collaboration of all authors. D.K. wrote the paper; J.-W.P. and S.H.L. supervised the paper. All authors have read and agreed to the published version of the manuscript.

**Funding:** This work was supported in part by the National Research Foundation of Korea (NRF) (Grant number: 2020R1A3B2079407), the Ministry of Science and ICT (MSIT), Korea and in part by the Korea Institute of Energy Technology Evaluation and Planning (KETEP) grant funded by the Korean government (MOTIE) (Grant number: 20192010107050).

**Conflicts of Interest:** The authors declare no conflict of interest.

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
