# Peer review of "A Study on the Power Reserve of Distributed Generators Based on Power Sensitivity Analysis in a Large-Scale Power System"

_electronics, doi:10.3390/electronics10070769_

Round 1

Reviewer 1 Report

I'm very sorry, but I wasn't able to understand your English, thus there is no possibility to point more detailed comments.

Your work should be more thoughtful in terms of cause-effect and logic. The described problems and considerations should arise from each other and their presentation should be clearer.

Author Response

Thanks for the comments. In response to the reviewer's comment, the authors improved the quality of English sentences through MDPI's English service. Also, the authors have fully addressed the comments raised by reviewers. Please see the authors’ responses to the reviewers’ comments from the “Track Changes” function in Microsoft Word.

Reviewer 2 Report

The authors carried out the study of VMS droop control model for power reserve analysis. This study showed several case studies to verify the effectiveness of the proposed model. This manuscript is well-organized and supports the conclusion. I would recommend this manuscript for publication with a few minor suggestions.

  1. What is the level of energy loss across the models?
  2. Is the VMS droop control model can be used (or applied) to existing infrastructures?
  3. Discussing any challenges or issues to be overcome in the conclusion section would be beneficial to readers.

Author Response

Thanks for the comment. Please see the attached author's response file.

Reviewer 3 Report

The paper looks good, it has appropriate citations and references. The technique pursued in the paper is good, the results show good improvement. The only concern with the results is there is no validation for the model. If you could address this concern with previous literature. It would be valuable for the research community.

Author Response

[A] Thanks for the comment. Due to the reviewer’s suggestion, the reference literature [17] was newly added in line 79. It shows that the power imbalances caused by changes of generation or load demand were adaptively shared by the VMS droop control. In this paper, we applied the VMS droop control to a large-scale power system on a national scale, and the concept of appropriate power reserve of CBGs required for the control was proposed.

Also some comments were newly added in conclusion section, such as lines 289 to 291. The power system in simulations of this paper was the real Korean power system based on the practical data. Using this actual data for the Korean power system, the power sensitivity and power reserve were calculated with sufficient reliability.

Reviewer 4 Report

The article presents a modern and highly appreciable contribution in the filed. A particular emphasis on the Korean government strategy within the "Renewable Energy 3020 plan" is of utmost interest to colleagues all over the world. The proposed solution for the inertia constant increment of a power system and its enhanced stability versus changing load presented clearly and comprehensively. 

The reviewer suggests the following reference be included:

Kim, J.H., Kim, S.Y. and Yoo, S.H., 2020. Public acceptance of the “Renewable Energy 3020 Plan”: evidence from a contingent valuation study in South Korea. Sustainability, 12(8), p.3151.

And magnify the scale of figures 3, 4, 7, 8.

As to the rest, the quality of the article is high enough to consider it for publication "as it is"

Author Response

[A1] Thanks for the comment. As suggested by the reviewer, the suggested reference was added as reference literature [3] in the paper. And lines 44 and 45 in the introduction section were added for this.

[A2] Thanks for the comment. As suggested by the reviewer, figures 3, 4, 7, 8 were replaced with magnified figures 3, 4, 8, 9 to help readers understand. Due to the new addition of figure 6, figures 8 and 9 were the existing figures 7 and 8.

Round 2

Reviewer 1 Report

Figures could be improved, resolution is very low and their sizes differ